# Real-Time Wireless Platform for In Vivo Monitoring of Bone Regeneration

**DOI:** 10.3390/s20164591

**Published:** 2020-08-15

**Authors:** Pablo Blázquez-Carmona, Manuel Sanchez-Raya, Juan Mora-Macías, Juan Antonio Gómez-Galán, Jaime Domínguez, Esther Reina-Romo

**Affiliations:** 1Escuela Técnica Superior de Ingeniería, Universidad de Sevilla, 41092 Seville, Spain; jaime@us.es (J.D.); erreina@us.es (E.R.-R.); 2Escuela Técnica Superior de Ingeniería, Universidad de Huelva, 21007 Huelva, Spain; msraya@diesia.uhu.es (M.S.-R.); juan.mora@dimme.uhu.es (J.M.-M.); jgalan@diesia.uhu.es (J.A.G.-G.)

**Keywords:** external fixator, wireless acquisition system, load sensor, bone regeneration, calibration, callus stiffness

## Abstract

For the monitoring of bone regeneration processes, the instrumentation of the fixation is an increasingly common technique to indirectly measure the evolution of bone formation instead of ex vivo measurements or traditional in vivo techniques, such as X-ray or visual review. A versatile instrumented external fixator capable of adapting to multiple bone regeneration processes was designed, as well as a wireless acquisition system for the data collection. The design and implementation of the overall architecture of such a system is described in this work, including the hardware, firmware, and mechanical components. The measurements are conditioned and subsequently sent to a PC via wireless communication to be in vivo displayed and analyzed using a developed real-time monitoring application. Moreover, a model for the in vivo estimation of the bone callus stiffness from collected data was defined. This model was validated in vitro using elastic springs, reporting promising results with respect to previous equipment, with average errors and uncertainties below 6.7% and 14.04%. The devices were also validated in vivo performing a bone lengthening treatment on a sheep metatarsus. The resulting system allowed the in vivo mechanical characterization of the bone callus during experimentation, providing a low-cost, simple, and highly reliable solution.

## 1. Introduction

Regeneration processes are intrinsic mechanisms in bone tissue that commonly appear along human lifetime, during fracture healing, bone remodeling, or growth in children. Depending on the process, bone regeneration may have different goals: to recover the skeletal functions of the body, to renew bone tissue, or to repair defects. Bone regeneration is carried out through multiple key factors; not only biological ones are important [1], but also the mechanical ones. Mesenchymal stem cells (MSCs) are considered bone progenitor cells with the ability to differentiate into osteoblasts, the main source of bone tissue [2]. The fate of these MSCs and the type and quality of the regenerated tissue are extensively proven to be sensitive to the mechanical environment, including the loading conditions, the stress distribution, or the fixation stiffness [3].

In orthopedic, oral, or maxillofacial surgery, bone regeneration processes are commonly applied in the treatment of several bony pathologies, e.g., bone disparities, skeletal reconstructions, or hemifacial microsomia [4,5]. Each clinical case requires specific treatments, e.g., fracture healing, distraction osteogenesis, or tissue engineering. In recent years, an extensive literature has developed on novel technologies to increase the regeneration rates and efficiency during clinical procedures [6,7,8,9]. For instance, Hatefi et al. [6] developed a distraction device based on a lead screw translation mechanism to support a continuous distraction of several craniomaxillofacial areas. A novel acoustic signal system, which employs ultrasound shear stress, was recently designed and tested in vivo by Machado et al. [7] to reduce bone fracture healing time. Regarding tissue engineering, the research is mainly focused on the optimization and manufacture of scaffolds or implants, which guarantee the cell infiltration and differentiation, such as the development of nanocomposite magnetic scaffolds carried out by Russo et al. [8] or the hybrid HA/PCL coaxial scaffolds manufactured by robocasting by Paredes et al. [9].

In the monitoring of these bone regeneration treatments, the widely used standard ex vivo techniques, such as micro-CT, SPECT, or histological studies [10,11,12], provide bone regeneration rates and hydroxyapatite concentration with high accuracy. Nevertheless, these techniques require the slaughter of the animal at specific time-points after surgery. The traditional in vivo clinical evaluations, such as plain film radiology or the review of the mobility of a fracture, are not quantitatively accurate enough to assess the evolution of the mechanical properties during the whole regeneration process [13,14,15]. CT scans are also used as a noninvasive technique to monitor skeletal diseases, but their radiation doses or the refraction of metallic fixations discard it as a proper method for continuous assessment of long-term bone regeneration processes. Moreover, newer non-invasive techniques have been proposed in the last few years, such as osteointegration monitoring based on ultrasound [16]. However, still, as far as the authors are concerned, they have not already been implemented in vivo.

In the field of orthopedic treatments of limb bones, in vivo mechanical experiments predominantly make use of fixations to stabilize the defect or the scaffold internally or externally, ensure the correct bone alignment, and avoid frequent bending problems in order to recover mechanical functions as fast as possible [17]. This device is commonly used as a less invasive technique in fracture fixation, deformity correction, limb lengthening, or treatments of defects. Along with this device, several biomechanical techniques are implemented to directly monitor the evolution of the mechanical properties of the new bone tissue. For instance, acoustic emission studies are based on analyzing the load required to initiate an acoustic response [18]. Vibrational studies similarly quantify the tissue regeneration from resonant frequency and wave propagation analysis [19,20,21]. The main disadvantage of these techniques is their great dependence on their experimental protocol, which is an obstacle to their standardization [22].

As an alternative, some studies have directly instrumented the external fixator for monitoring the bone regeneration processes quantitatively [23,24,25,26]. Loads, displacements, accelerations, or strains are measured by a wide variety of sensors, which are coupled with an acquisition system to assess the bone callus stiffness in vivo. Conventional strain gauges are typically used in unilateral fixators to obtain easy post-processing sagittal and frontal strains [27,28]. Nevertheless, the lack of endurance of these sensors, their low durability, and their arduous gluing tasks and protections against impacts make them unfeasible for long-term animal experimentation. Accelerometers are also used to carry out in vivo impact testing in order to estimate the frequency response function and the resonant frequencies of the bone during healing. Despite its reduced cost, this technique is limited to experimentation at rest and is not sufficiently tested in complex bone regeneration processes [29]. Load cells are other common sensors used for monitoring forces through the external fixator over the experimental phase. Depending on its specifications, its insulation is necessary for facing bending moments, which complicates its embedding in the fixator and the continuity of the in vivo measurements in break situation [30]. These measurements allow identifying the time-points when the new tissue is consolidated enough to proceed with the external fixation removal [25]. Furthermore, previous studies in the literature have also proposed some of the mentioned techniques in cases of internal fixation. On the one hand, Tan et al. [31] developed a sensor for bone plate strain monitoring. On the other hand, Chiu et al. [32] proposed a technique to assess the stiffness of an internally fixated femur based on its frequency response. However, the implementation of these sensors in internal fixations are restricted to bone or fracture healing processes. In addition, as far as the authors know, none of these techniques have been tested in vivo.

Unlike studies with small size specimens [33], the portability of these electronic systems is a relevant specification in animal models with human-like bones in order to avoid disturbing their natural gait. Previous studies either used acquisition systems that are not portable [34,35] or did not provide data about portability [33,36]. In the literature, the control system hardware is usually integrated by multiple commercial devices fixed to a structure, a hardware board [27,34,36], considerably increasing their full size and energy consumption. For instance, Reifenrath et al. [34] needed a signal amplifier composed of a battery, ports, and wireless transmission for their bending strain measures in sheep tibiae. Likewise, the acquisition system used by Meyers et al. [36] in their distraction measures is constituted by multiple commercial devices fixed to a wooden board: power supply, DAQ, sensor amplifier, or motor driver. Furthermore, in some biomechanical in vivo studies, the volume of monitored data is input manually into the computer [33]. This labor-intensive procedure is prone to potential transcript errors, highlighting the importance of the automation of the data acquisition process. The data receiver and manual functions are also commonly controlled by commercial paid software [24,28,30,33,34,35]. Besides saving costs per license, the development and implementation of free software would allow adapting the data storage and its remote analysis to the specific measure. Finally, the robustness of the packaging is also an important point in the design of an acquisition system for experiments with animals due to their unpredictable behavior. In the literature, robustness data are not usually provided [27,30,33,34,36].

The aim of this work is to design, manufacture, and validate an external instrumented fixator that allows performing a mechanical monitoring of several bone regeneration processes of the lower limb bones (e.g., femur, tibia, or ovine metatarsus), improving the characteristics and the accuracy of previous devices. From the electronic point of view, the main objective of the work is to design a reproducible, small-size, low-cost, battery-operated, microcontroller-based, and autonomous data acquisition system, which easy to manipulate and avoids the loss of information during the experimental measurements. These devices will be calibrated ex vivo and tested in vivo for a bone lengthening treatment in a sheep.

## 2. Materials and Methods

The designs of both the external fixator and acquisition system have the main feature of being handy, versatile, and customizable for different bone regeneration applications, e.g., fracture healing, distraction osteogenesis, or tissue engineering. Their simultaneous operation allows collecting data from the sensors and indirectly monitoring the progress of the clinical treatment. The sensor network is made up of a combination of several technologies, including hardware, software, and firmware. All mechanical and electronic devices, as well as their calibration are detailed below.

### 2.1. Distractor, Mechanical Design

An external fixator, which is shown in Figure 1A, was designed for the stabilization and immobilization of long bone fractures and other bone defects in sheep. The design must be strong enough to resist biomechanical forces and flexible enough to allow mechanical stimulation in the bone defect [37]. The fixator was mainly composed of two stainless steel frames clamped to the bone fragments by means of three Schanz pins per frame and interconnected by four bars. Two models of bars were designed for multiple bone regeneration processes. Extendable bars (Figure 1B) allow the controlled displacement of one bone fragment by using a nut-screw mechanism, which is used in treatments that imply elongation, including bone lengthening, bone shortening, or bone transport. Once the elongation is finished, the displacement is prevented using small screwed pieces, which inhibit the rotation of the nut. Fixed bars (Figure 1C) were also designed for bone regeneration processes that do not require an elongation process such as fracture healing, bone consolidation, or tissue engineering. Both bars were instrumented with load sensors that allowed the force’s acquisition throughout the fixator during the experimental tests. Durability and ease of assembly were noteworthy points in the selection of these sensors. Burster^®^ 8431-6001 load cells (Burster, Gernsbach, Germany) were chosen, sensors that tolerate bending moments and avoid including complex isolation mechanisms in the bars. Moreover, their measurement range is 0–1 kN, enough range to bear the gait load of most of the typical animals under experimentation. For instance, the average highest forces measured on Merino sheep metatarsus (internal forces) were 1.26 times the total weight of the animals in a bone transport study in the literature [38,39]. Assuming an average sheep body weight of 53.5 kg [38,39], the internal force would only reach around 661 N distributed among the four load sensors of the fixation.

### 2.2. Hardware Design

The hardware of the acquisition system was designed according to the needs of the animal experimentation. All measurements parameters were collected by a control unit, whose processor is an ESP32^®^ (Espressif Systems, Shanghai, China). This control unit integrates a main microprocessor (32-bit LX6, Tensilica Xtensa^®^, Tensilica, Santa Clara, CA, USA) and an ultra-low power co-processor, as well as peripheral interfaces. A block diagram of the whole acquisition system is shown in Figure 2.

Two central systems constitute the functional structure of the acquisition system. On the one hand, the Power Supply system provides power support for the device operation. On the other hand, the load cells assembled to the external fixator require a Measurement System for the signal conditioning and the A/D conversion. Both main systems are described in the following Section 2.2.1 and Section 2.2.2. Moreover, a real-time clock (based on DS3231 RTC) maintains the temporal synchronization of the processor so as to provide a timestamp to every measurement block taken by the system. Finally, a memory card is used as a local data storage. Once an experimental test is finished, the data can be accessed by the monitoring application or removing the memory card. This application, which was made in Python, allows the in vivo monitoring of the proper development of the test and the subsequent analysis of the collected data. More details of the monitoring application are specified in Appendix A.

Figure 3 shows the details of the hardware implementation of the data acquisition system and the packaging designed to protect it. The final dimensions of the sensor packing were 125 × 80 × 33 mm. A carefully designed layout was required to reduce the conducted noise, isolating the signal conditioning and A/D conversion from the rest of the components.

#### 2.2.1. Power Supply System

A scheme of the Power Supply system is detailed in Figure 4. It is composed by a LiPo battery of 1.5 Ah and 3.7 V, as well as a battery charging circuit, which avoids introducing an electricity grid connection. The LiPo battery ensures the connectivity for each experimental test providing enough energy for 15 h. An additional circuitry was included based on the integrated linear charger TP4056 as a system power-path management device, which can work with micro-USB and fixes the charge voltage at 4.2 V. A FS312F-G battery circuit protection together with integrated transistors FS8205A were also added.

From the battery voltage, two other voltages were generated using the AS1117-3.3 low dropout regulator and the ME2149 step-up switching regulator: one of 5 V required for generating the reference voltage of 2.5 V of the four Wheatstone bridges of the load cells and the second one of 3.3 V to supply the rest of the system components. Additional functionality was added to enable the system to be turned on by pressing a button and off by a subsequent press. Using the ON-OFF signal through the R1 resistor, the microcontroller detects the pulse as a zero logical level, taking transistors M1 and Q1 to the cut-off region, which causes the CEsignal of the ME2149 regulator (Figure 4) to be zero, turning it off. The firmware code was modified so that all disk write operations are finished before the power is cut.

#### 2.2.2. Signal Conditioning and A/D Conversion

The four load cells of the external fixator are connected to the hardware board through 6-pin mini-DIN connectors. The operating voltage of these sensors is 2.5 V, the reference voltage during calibration, which must be kept constant during experimentation. For this purpose, a Reference voltage generator circuit was implemented (Figure 5 left). It consists of the REF3025 voltage reference, which offers high precision, low dropout voltage, small size, and low power consumption. A positive feedback loop around the AD8542 amplifier was added to maintain the 2.5 V voltage over the entire current range demanded by each load cell. Besides, the loop includes protection against short circuits in load cells through resistor R2 and transistors Q2, Q3, and Q4.

The measurements of the load cells are conditioned by means of the instrumentation amplifier INA2126. It features low noise, precision, a low quiescent current, and a wide operating voltage range, making it ideal for portable instrumentation. This amplifier was operated from a single power supply with careful attention to input common-mode range, the output voltage swing of both op-amps, and the voltage applied to the reference (REF) terminal. This fact facilitates the design of a battery-powered system. The output REF pin is used to level shift the internal output voltage into a linear operating condition. This pin was connected to a potential, which is the mid-supply of the 2.5 V, avoiding saturating the output of the amplifiers. The output of the INA2126 instrumentation amplifier is directly connected to the external ADS1115 A/D converter (Figure 5, right) from Texas Instruments. This converter was chosen because it offers a 16-bit accuracy above the 10-bit of the internal converter of the ESP32. The ADS1115 features an input multiplexer, which allows two differential or four single-ended input measurements. Two of these converters, operating in differential mode, were included to raise the conversion speed and further reduce the conducted noise. The differential mode allows for measuring both negative forces (compression) and positive forces (traction) to adapt the system to static and dynamic measures in any bone regeneration treatment.

The ADS1115 A/D converter can operate in either continuous-conversion mode or single-shot mode. The device is automatically powered down after one conversion in single-shot mode, reducing the power consumption significantly. The shot mode was selected so that the number of measurements to be acquired (from 5 to 50 per second) can be controlled thanks to the incorporation of the real-time clock. Concerning sampling, the measurements must be carried out in fixed time intervals to analyze the footsteps and ensure data synchronization with other measuring devices, such as a load platform. The real-time clock ensures the correct timing even in situations with a lack of WiFi signal.

### 2.3. Firmware

The ESP32 microcontroller used has two processors with shared memory. The firmware was programmed using FreeRTOS, a real-time operating system (RTOS). The system allows users to execute several tasks simultaneously, distributing them between both processors. Delays in the execution of tasks, the memory fragmentation, or the possible data corruption were avoided by means of the static memory allocation process. The main system modules involved in the firmware are (see Figure 6):*Main*: the central program responsible for initializing system tasks and leaving it in standby mode. Before calling the main program, it executes the initial configuration and other internal tasks responsible for the maintenance of the system and other parts of the ESP32 (WiFi system, internal storage system, or the run-time support).*Logger*: This module ensures the correct time and date at startup employing the real-time clock (DS3231 RTC), which has its own battery. In case of incorrect data, it attempts to connect to WiFi to retrieve the time information from a time server using the Network Time Protocol (NTP). The module also writes and reads collected data and diagnostic messages to the micro-SD memory. This storage on the micro-SD card is independent of the wireless transmission, ensuring the proper collection of the in vivo measures in cases of the loss of WiFi signal.*WiFi*: ensures the connections and the correct operation during experimentation, operating as a client or as an access point (AP).*Server*: This module, which was developed using the *netconn* library of the ESP32 SDK, activates and waits for clients to connect. Once the client is connected, the access to a command interpreter is available, allowing making calls to different system functions: acquisition, capture, or reading configuration and information download from the disk.*Capture*: performs the process of reading the sensors, integrating two synchronized tasks: the data capture task and the data storing task. The two A/D accessible via the I2C port are used so that, every 2 readings, a signal multiplexing is performed to change the capture channel.

The data transmitted and stored must include an adjustment or normalization due to a drift produced by the sensors themselves, by temperature, or some constant force applied to the external fixator (for example, gait forces). Before the beginning of the experimental test, a number of measurements are initially taken in a static position of the limb of the animal, and the arithmetic mean is calculated. Therefore, a normalization value is subtracted from the measurements of each load cell. Once the normalization values have been determined, the four measurements of each load cell are stored next to a three-byte mark that ensures the synchronization of the receiver in case of data loss.

After the acquisition of 10 samples in a 150-byte buffer, the acquisition task goes on working on another buffer while previous data are sent. Therefore, the acquisition system works in real time. Considering that the ESP32 contains two 32-bit processors working at 200 MHz, the processor load is around 20%, which is quite reduced and results in low power consumption. Moreover, the data readings were programmed using the non-blocking operation mode. In this mode, if there are no data to read, the call will return immediately, preventing the tasks using the socket (WiFi network) from being blocked. Their main advantages against the blocking mode are the fluency of the device, the ability to stop the acquisition at any time, and the storage disruption in cases of writing errors. Figure 7 shows flowcharts of both the data acquisition processes (Figure 7A) and the data reading operation mode (Figure 7B).

### 2.4. Force Measurements

In the treatment of long bone pathologies, there are some processes that keep the length of the limb (e.g., bone transport, bone healing, or tissue engineering [30,38,40]) during the regeneration process and others that imply its modification (e.g., bone shortening or lengthening [41,42]). The selection of the proper treatment depends on the clinical pathology of the patient. Some cases require the controlled displacement of one bone fragment by means of the external fixator up to the distance imposed by the specific treatment (distraction phase) [26,43]. Once the bone fragment position is fixed, the subsequent stage of mineralization of the bone callus (consolidation phase) begins until the complete tissue remodeling. Treatments without displacement of the bony fragment are exclusively composed by the consolidation phase. The measures and devices necessary to assess the biomechanical evolution of the bone regeneration depend on the phases that occur.

*Distraction measures*: After applying a displacement of a bony fragment, the reaction force of hard and soft tissues to distraction is measured by means of the external fixator and the acquisition system ($F_{f}$) at rest [26,44]. Assuming the absence of movement in the treated limb, the monitored force corresponds to the traction force applied on the bone callus ($F_{c}$) for its axial deformation.*Consolidation measures*: Gait analysis is a common non-invasive technique that allows quantitatively assessing the evolution of multiple bone pathologies during the consolidation phase [25,38,45]. In bone regeneration processes, these measurements require monitoring the forces through an instrumented fixator and the ground reaction force (GRF) during the steps of the animal [25,38]. The GRF, which is commonly quantified by a load platform (Figure 8A), is an important input in biomechanical analysis and represents a part of the internal force through the skeletal structure of the animal ($F_{a}$). Muscles and soft tissues store the rest of the internal force during a stance phase, and this is not directly quantifiable. In the operated limb, forces through the skeletal structure ($F_{a}$) are divided between the external fixator ($F_{f}$) and the bone callus ($F_{c}$) depending on its degree of mineralization (Figure 8B). Therefore, the load through the bone callus was calculated from both previous loads using Equation (Equation 1).
(1)$$F_{c}=F_{a}-F_{f}$$

### 2.5. Models for Stiffness Estimation

Monitoring the stiffness of the bone callus during most of the bone regeneration processes (e.g., distraction osteogenesis, fracture healing, or tissue engineering) requires discriminating between two estimation procedures, depending on the clinical phase: distraction or consolidation.

On the one hand, the evolution of stiffness during the distraction phase is easily calculated from the daily controlled displacement applied to the bony fragment (▵d) and the monitored axial force ($F_{f}$) using Equation (Equation 2) (during distraction measures $F_{a}$ = 0 and $F_{f}$ = $F_{c}$).
(2)$$K_{c}=\frac{F_{f}}{▵d}=\frac{F_{c}}{▵d}$$

On the other hand, an analytical model of the bone-fixator system was defined to estimate the stiffening of the bone callus during the consolidation phase (Figure 8). The instantaneous callus stiffness value ($K_{c}$) was estimated from the forces through the skeletal structure of the limb under treatment ($F_{a}$), the forces through the load cells of the external fixator ($F_{f}$), and its stiffness ($K_{f}$), taking into account the stiffness of the bars, frames, and flexion of the screwed pins from Figure 1A. The callus stiffness was previously measured performing compression tests in the push-pull machine, obtaining a mean value of 593 ± 21 N/mm. Disregarding the strains of the cortical bone compared with the bone callus strains, the stiffness of the bone callus ($K_{c}$) was calculated by Equation (Equation 3).
(3)$$K_{c}=K_{f}\cdot \frac{F_{c}}{F_{f}}=K_{f}\cdot \frac{F_{a}-F_{f}}{F_{f}}$$

### 2.6. In Vitro Calibration

Force acquisition system calibration was carried out by performing in vitro tests in a metatarsus of a Merino sheep using a push-pull testing machine MTS 858 MINIBIONIX^®^ II (MTS System Corporation, Eden Prairie, Minnesota, USA). An ex vivo surgery was previously performed in the laboratory where the instrumented external fixator was assembled to the metatarsus extracted from a cadaveric sheep. Additionally, an epoxy resin, EpoxiCure^®^ 2 20-3430-064 (Buehler, Esslingen, Germany), was added at both distal and proximal ends of the bone, improving the hold inside the test machine. The acquired force slightly exceeded −661 N, taking this value as an upper bound of internal force experienced in bone regeneration treatments in sheep according to the literature [26,38,46,47]. The error in the acquisition was calculated by comparing forces measured by the load cells with those registered by the push-pull machine during compression tests. Furthermore, noise in the measurement signals was quantified by the standard deviation in an empty signal with zero loads.

Likewise, the model for callus stiffness estimation presented in Section 2.5 was also tested in vitro by means of elastic springs with different stiffness ($K_{sr}$) following the scheme presented in Figure 9. Assuming uniform mechanical properties throughout the complete bone callus, the spring represents its stiffness at different time-points of the experimentation process: between 16.75 and 208.83 N/mm for distraction tests and between 103.01 and 7448.78 N/mm for consolidation tests [25,26,44,48]. In distraction tests, reaction forces from the distraction of bony fragments ($F_{f}$) were simulated with decompression (1 mm) of the springs, obtaining the estimated stiffness ($K_{s}$) directly using Equation (Equation 2). In consolidation tests, forces through the bone ($F_{a}$) were directly applied by the compression test machine, and the stiffness of the springs estimated from the bone-fixator model ($K_{s}$) was calculated by applying Equation (Equation 3). Both distraction and consolidation estimations of stiffness were compared with the reference stiffness value ($K_{sr}$) obtained from an average of 5 direct tests on the springs. The relative error and uncertainty for each estimation were calculated, taking into account several sources of uncertainty: the reference spring stiffness uncertainty, the replication uncertainty between bone-fixator assemblies, and the repetition uncertainty associated with identical measures for the same assembly. For each elastic spring and experimental phase, a total of 4 different assemblies of the external fixator (replication error) were performed, and 5 stiffness estimations (repetition error) were calculated for each assembly. More details of the uncertainties calculation are presented in Appendix B. The elastic-linear behavior of each spring was previously checked.

### 2.7. In Vivo Measurements

The applicability and the operation in vivo of these new devices were tested by means of a bone lengthening treatment, an example of a bone regeneration process that involves the modification of the limb length. This regeneration process was performed in the right-back metatarsus of a female Merino sheep (67 kg). This animal model was selected due to its docility and the reported similarities in long bone mineral composition, remodeling rates, and dimensions to humans ones [49,50,51]. Besides, the adjacent soft tissues around this metatarsal bone are solely the tendons of deep and superficial digital flexor and the tendons of long digital and lateral extensor, which simplifies the surgery compared with other ovine bony models with surrounding muscles [52]. The authorization of the Experimentation Ethics Committee of the University of Seville was obtained to carry out the experiments. Figure 10 shows the implantation of the designed external fixator in the sheep metatarsus and the performance of the osteotomy during surgery, which divided the long bone into two bony fragments. The clinical protocol consisted of a latency period of one week, a distraction phase of 15 days with a distraction rate of 1 mm/day, and a consolidation phase until the slaughter of the animal. Non-instrumented interchangeable bars were initially assembled in the fixation during the surgery and latency period to avoid the deterioration of the sensors during unmeasured phases. These bars were exchanged for the designed instrumented ones (Figure 1) before the beginning of the distraction process.

The bone callus stiffness was measured in vivo at different time-points of the distraction and consolidation phases. The distraction tests were carried out with the animal laid on the floor using the extendable bars exclusively. These bars allow with their crew-nut system separating the bone fragments the defined distraction rate in a controlled way. The stiffness value obtained from distraction tests is the instantaneous “peak callus stiffness” (PCS) of the corresponding distraction day due to the viscoelastic behavior of the callus tissue. The system operation during distraction measures was tested in vivo on Days 1, 6, 9, and 12 of the distraction phase. Meanwhile, the consolidation measurements were carried out using load cells and a load platform, Pasco PS-2141^®^ (PASCO, Roseville, CA, USA), embedded in a walking gait, which measures the GRF from the footsteps of the sheep. For these measurements, it was assumed that the relation *$F_{a}$/GRF* is constant with a value of 3.22 during the complete consolidation phase [30]. The increase in the stiffness of the bone callus of the sheep was estimated in vivo through the model presented in Section 2.5 at different time-points: Days 3, 8, 10, and 12 of the consolidation phase.

Additionally, the noise of the acquisition in vivo was also quantified by the standard deviation during measures at rest with the sheep lying.

## 3. Results

### 3.1. In Vitro Results

Compression forces applied by the pull-push machine to the metatarsus (red signal) and forces measured by the external fixator assembled to the bone (blue signal) are shown in Figure 11. The relative error in the force acquisition (yellow signal) was calculated taking the push-pull machine signal as the reference force values during the testing time. The relative error slightly increases throughout the compression test. However, it averages 3.66% and barely exceeds 4% at the end of the calibration test.

The noise level during in vitro force measurements also reveals promising results. The average noise level of each sensor scarcely reaches 0.18 N, while this value only rises up 0.36 N in the complete external fixator. Nevertheless, the noise increased under in vivo boundary conditions, as presented in the following section.

The stiffness of the different elastic springs used to validate the stiffness model ($K_{sr}$), their mean estimated values ($K_{s}$), and their standard deviation ($\sigma_{s}$) are shown in Table 1. Furthermore, the error (*e*) and uncertainty values (*U*) of each spring (from the estimations detailed in Appendix B) are also presented. Overall, the errors and uncertainties do not show a direct relationship with the degree of callus mineralization. While the mean relative error (*e*) is similar between both experimental phases, 6.75% for distraction and 4.90% for consolidation, the mean uncertainty value is slightly higher in consolidation tests, 14.04%.

### 3.2. In Vivo Validation

During distraction tests, axial forces were measured for 20 min from the beginning of distraction. The mean noise in the in vivo force measurements reached 3.01 N due to the experimentation boundary conditions. Figure 12A shows an example of a measure of the ninth day of distraction. The viscoelastic behavior of the bone callus tissue is noticeable after undergoing the deformation produced by the separation of the bony fragments. The instantaneous peak force is initially monitored, as well as the later relaxation of the tissues during the whole in vivo test.

The PCS at the time-points specified above is shown in Figure 12B, as well as their uncertainty range according to the mean value calculated in vitro. The stiffness of the bone callus increases over the days, reaching about 250 N/mm at the latest days of the distraction phase. The stiffening of the bone callus on the tested days of the consolidation phase with its corresponding uncertainty range is also presented in Figure 12C. The results suggest exponential mineralization of the bone callus, increasing its stiffness from 0.53 kN/mm to 1.85 kN/mm in 30 days of consolidation.

## 4. Discussion and Conclusions

The main purpose of this study is to present and validate a fixator capable of adapting to multiple bone regeneration processes, accompanied by an acquisition system optimized for animal experimentation. From the mechanical perspective, most authors in the literature experiment with manufactured fixations, the designs of which focus exclusively on their specific application. Despite the technological advances in the induction of electromechanical stimulations, most of the reported unilateral fixators present particular mechanical designs for distraction osteogenesis applications [36,47]. In the same way, the telemetric internal fixation of Kienast et al. [53] and the instrumented external fixators designed by Grasa et al. [27] and Reifenrath et al. [34] concentrated on the stabilization and in vivo assessment of bone or fracture healing. Their inability to apply distraction prevents their adaptability to a certain amount of processes, including bone lengthening, shortening, or bone transport.

Meanwhile, the external fixator contrived by Claes et al. [54] and Mora-Macías et al. [30] has a specific design for bone transport treatments in sheep metatarsus. Moreover, the design of Claes et al. [54] is limited to interfragmentary movement measures due to its instrumentation based on displacement transducers. Contrastingly, direct force measurements and indirect callus stiffness estimations were carried out by means of the mechanical design of Mora-Macías et al. [30]. Nevertheless, force measures in bone transport requires including multiple sensors so as to discriminate the mechanical behavior of different soft tissues (the bone callus and the docking site). This fact and the need to include mechanical isolation for their selected sensors against bending moments make the design of Mora-Macías et al. [30] complex for other bone regeneration processes.

Contrary to the previous devices, the designed external fixator of this study delivers significantly better versatility to adjust to different treatments in the same specimen. The combination of extensible and fixed bars permits assembling an external fixator adapted to the characteristic of each clinical case and the investigation of several bone healing models. For instance, different stages of a fracture healing model could be assessed, including the loss of bearing capacity because of the characteristic inflammatory response, the mechanical properties of the resulting hematoma, and its recovery and stiffening during the soft tissue formation, ossification, and remodeling [20,55]. Likewise, mechanical tension-stress forces from the gradual traction of living tissue (bone callus tissue after latency and surrounding soft tissues if it implies limb elongation) could be quantified during the distraction phase of bone lengthening and bone transport treatments, as shown in the preliminary in vivo results of the manuscript [26,36,56]. Like fracture healing, the increase of the elastic modulus of the bone callus and bearing capacity in the treated limb could be monitored during the consolidation phase [25,38]. In the field of tissue engineering on large bone defects, the external fixator could be a potential tool for the stabilization of patient-specific scaffolds and bony fragments in critical-size defects. The influence of cell proliferation and bone tissue formation on the apparent mechanical properties of the scaffolds could also be measured by means of a bone-fixator model equivalent to that implemented in vivo, analyzing the influence of different materials and microstructures. As far as the authors are concerned, this last application has not yet been implemented in vivo. Nevertheless, the principal value of these devices lies in comparing mechanically, clinically, and histologically all of these bone regeneration models under a fixed mechanical environment and bony model.

Regarding the acquisition system, energetic efficiency, real-time performance, size, and sturdiness were the key factors in the design of the presented device. Table 2 shows comparatively the characteristics of acquisition systems for instrumented external fixations, revealing some unsatisfactory points for animal experimentation. Meyers et al. [36] reported electronic devices coupled with a wooden board exclusively for static tests (distraction and mechanical stimulation) due to the apparent volume, weight, and low protection of the devices. Grasa et al. [27], Reifenrath et al. [34] and Mora-Macías et al. [30] made use of devices with a real-time performance that were transportable enough for animal experimentation with sheep, but with room for improvements in dimensional terms. The portability, weight, and size are the primary benefits of the acquisition device used by Wee et al. [47], a miniature data logger. Nonetheless, this little device limits the measurement capacity to one reading per 15 s and does not allow real-time storage, which requires the periodic uploading of data to the computer and a delay in the data analysis. In addition, most of the devices mentioned above use commercial hardware and software, as shown in Table 2, increasing the volume, weight, and costs derived from their acquisition and adaptation to the requirements of the sensors in the experimental boundary conditions. To the best of our knowledge, none of these devices include a second real-time data storage, independent of wireless connectivity, and essential for animal experimentation during the gate (Table 2). The inclusion of a micro-SD storage, whose measurements can be downloaded from the monitoring application itself, represents a safe and functional solution to ensure clinical assessment. Therefore, beyond the implementation of specific hardware and software, the integrated acquisition device presented in this study is the most portable, reliable, and lightweight solution that incorporates real-time monitoring implemented in bone regeneration applications.

In terms of the quality of measurements, the average relative errors calculated in vitro were 6.73% during distraction and 4.90% during consolidation measures. These results improve those reported by Mora-Macías et al. [30] in bone transport: 7.8% during distraction measures and 9.5% during consolidation measures (Table 3). As far as the authors are concerned, no other previous instrumented fixation systems provided error data in their measurements [27,34,36,47]. However, the accuracy of our devices is also noteworthy when compared with other medical tools for bone regeneration processes, as presented in Table 3. The device for monitoring the bone callus torsional stiffness of Windhagen et al. [57] reported an average error of 15%; the complete device for bending stiffness measures during fracture healing of Hente et al. [58] quantified an error of 29.3%; and the orthometer designed by Eastaugh-Waring et al. [59] reported an average error of 10% monitoring the fracture stiffness. Few bone regeneration studies were found in the literature that report uncertainty levels in their measurements. Comparing with Mora-Macías et al. [30], our results show significantly lower levels of uncertainty in the distraction, 7.5% versus 22.6%, and consolidation phase, 14% versus 35.7%, suggesting minor reference, replication, and repetition errors in our measurements.

Unlike other recent internal fixation systems [31,32], our devices were tested in vivo on sheep metatarsus, whose preliminary results in bone lengthening (Figure 12B) agree with other studies in the literature. The peak stiffness of the bone callus estimated during the distraction phase rises as more elongation is applied, as observed in other results in the literature [26,44,48]. Forriol et al. [48], applying the same rate of distraction in lamb tibiae, recorded a bone callus peak stiffness after 13 mm of elongation slightly above 20 N/mm, a lower stiffness level compared with our results. The dimensions of the distracted bone were suggested to be the main source of dissimilarity. The peak stiffness results obtained by Mora-Macías et al. [26] in bone transport treatments in sheep were consistent with our results, a range of 50–150 N/mm on Day 13 of distraction. In this case, the lower peaks of stiffness (50–150 N/mm versus 220–250 N/mm) could be related to the lack of surrounding soft tissue elongation. Meanwhile, Meyers et al. [44] measured much more stable peak stiffness in sheep tibiae (approximately 110 N/mm) during the whole distraction phase. This fact may be a consequence of applying a lower rate of distraction (0.275 mm per distraction) and a higher daily frequency of distraction (two per day). Several biological factors are behind the measured stiffening of the new bone tissue. Firstly, the secretion of proteins, especially heteropolymer type I collagen, due to the hematoma formed during the latency period and the tension-stress distraction forces [60]. These collagen fibers along with other cells (chondrocyte-like cells and fibroblasts) are progressively oriented along the displacement axis and mature during the distraction phase [60,61].

By contrast, few studies have carried out in vivo monitoring of the stiffening process during the consolidation phase in bone regeneration processes. However, our preliminary results are in accordance with the stiffening behavior reported by Mora-Macías et al. [25] in bone transport on the same animal model. Comparing the same time-point, Day 36 from surgery, their estimated stiffness was around 2 kN/mm (versus 1.85 kN/mm in our preliminary results). The application of equivalent biomechanical factors, e.g., latency period and rate or frequency of distraction, is suggested to be behind these similar results. The evolution of the mechanical properties is suggested to be biologically caused by an increase in mineral density due to high recruitment and activity of osteoblasts [62]. A deeper bone lengthening study needs to be carried out to verify this preliminary conclusion.

Assuming a cross-section of the metatarsus (cortical bone and bone marrow) of 147.8 mm${}^{2}$ from geometries obtained after the segmentation of CT scans performed after the slaughter of the animal, the Young modulus of the bone callus varies between 0.16 and 21.01 MPa during the distraction phase and between 53.95 and 187.96 MPa during the days of the consolidation phase considered in this study. These preliminary results are consistent with the ex vivo fracture analysis of Leong and Morgan [63]. Indentation moduli of multiple tissue types was measured in their study: 0.61–1.27 MPa for granulation tissue, 1.39–4.42 MPa for chondroid tissue, and 26.92–1010 MPa for woven bone. Mora-Macías et al. [64] also measured an increase in the elastic modulus of the woven bone in several bones’ transport callus during the consolidation phase. According to these results, the variation of the mechanical properties of the bone callus could be mainly related to the increase of the woven bone concentration and maturation of woven bone during both the distraction and consolidation phases.

There are naturally some limitations in the present study. Regardless of the flexibility of the external fixator to be adapted to most of the regeneration processes, a second design of extendable bars would be necessary to adjust the fixator to bone transport treatments. These bars would allow the displacement of an intermediate bone fragment between two fixed ones. Another limitation involves the model for stiffness estimation. Firstly, the stiffness of the external fixator presents a significant dispersion, mainly due to the fixation between the bone and the Schanz pins. The high dependence of the model with such stiffness could distort the monitoring of the bone regeneration process in the daily clinical routine. Secondly, the model assumes a homogeneous behavior throughout the whole bone callus, neglecting the tissue heterogeneity found in previous ex vivo studies [43,44].

These findings provide a potential system for several bone regeneration processes in limb bones, adaptable for humans, easy to replicate, and electronically optimized. The devices and models permit the characterization of the bone callus under different healing models and biomechanical factors. Their versatility also allows a direct comparison between bone regeneration processes and a better understanding of their mechanobiology. The system also presents an improved accuracy compared to previous medical devices. Its portability, robustness, extra data storage, and real-time monitoring allow an exhaustive clinical follow-up without resorting to traditional techniques, including X-ray or manual inspection.

## Figures and Tables

**Figure 1 sensors-20-04591-f001:**
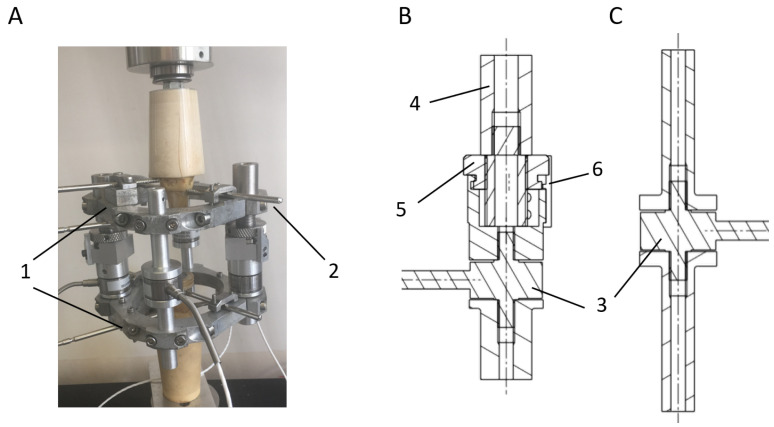
(**A**) Complete external fixator assembled to a sheep metatarsal bone: 1. stainless steel frames; 2. Schanz pins; (**B**) extendable bar scheme; (**C**) fixed bar scheme. Remarkable bars elements: 3. load cell; 4. screw for elongation treatments; 5. nut for controlling the displacement of the screw; 6. pieces for nut fixation.

**Figure 2 sensors-20-04591-f002:**
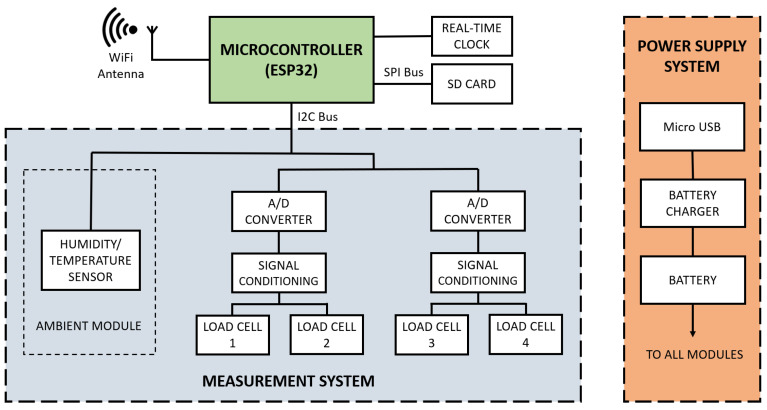
Block diagram of the data acquisition system: microcontroller, Measurement System (system parameter measurement), and Power Supply System.

**Figure 3 sensors-20-04591-f003:**
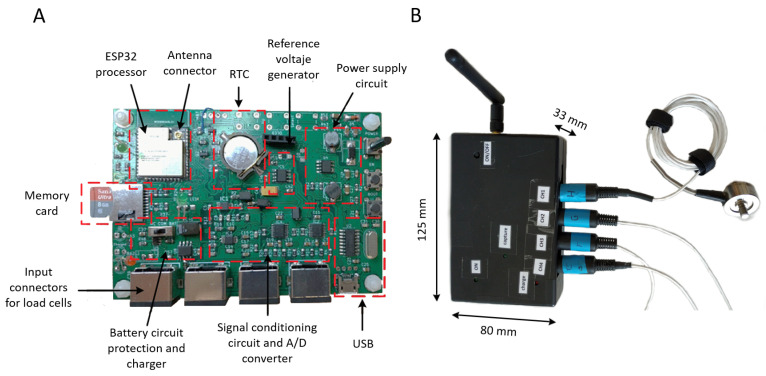
Image of the data acquisition system: (**A**) hardware board; (**B**) packaging design and its dimensions.

**Figure 4 sensors-20-04591-f004:**
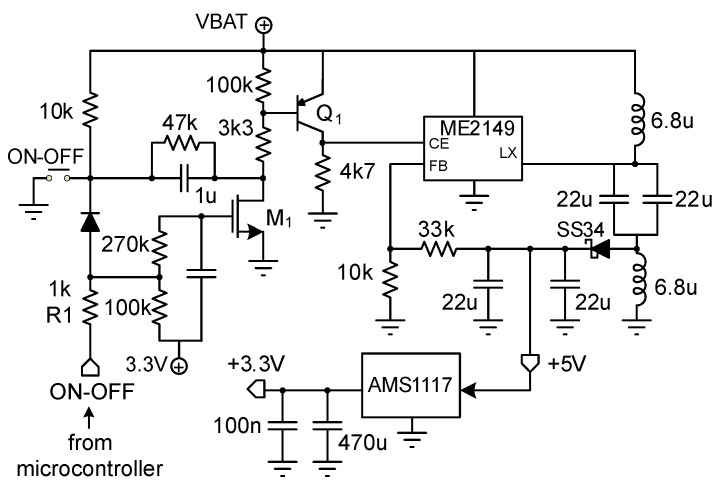
Power Supply System scheme.

**Figure 5 sensors-20-04591-f005:**
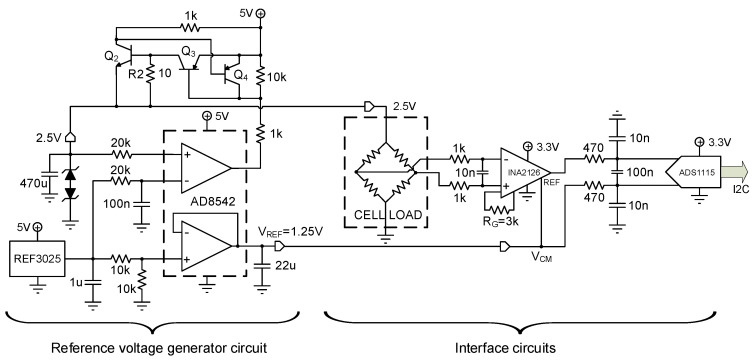
Signal conditioning circuit and A/D conversion: *Reference voltage generator circuit* (left) and *Interface circuit* (right).

**Figure 6 sensors-20-04591-f006:**
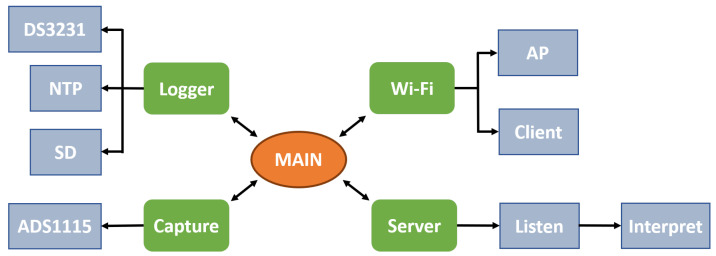
General block diagram of the firmware level operation and main modules: main, logger, capture, WiFi, and server.

**Figure 7 sensors-20-04591-f007:**
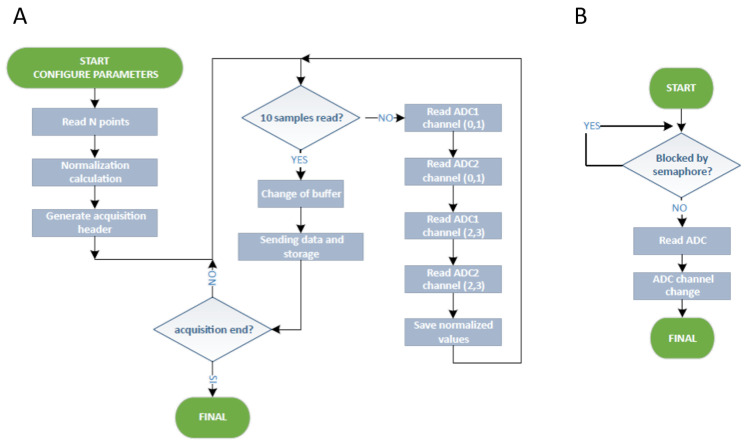
Flowchart of: (**A**) the data acquisition process; (**B**) data reading operation mode.

**Figure 8 sensors-20-04591-f008:**
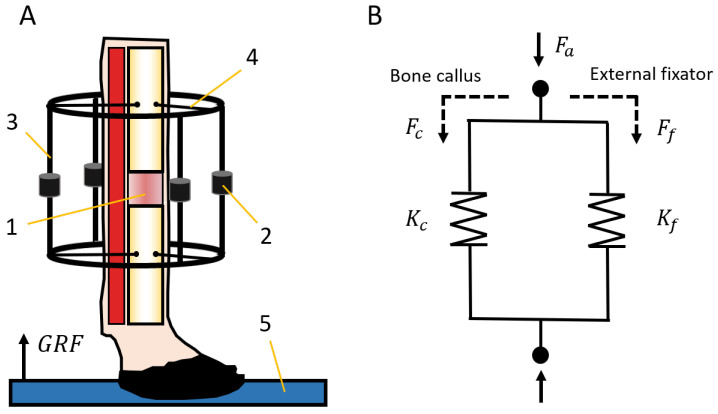
(**A**) Bone-fixator system scheme: 1. distraction callus from bone lengthening, 2. load cells, 3. external bars, 4. Schanz screws, 5. load platform. (**B**) Bone-fixator model: $F_{a}$, external load applied; $K_{f}$, stiffness of the external fixator; $F_{f}$, part of the external load through the fixator; $K_{c}$, stiffness of the distraction callus; $F_{c}$, part of the external load through the distraction callus. GRF, ground reaction force.

**Figure 9 sensors-20-04591-f009:**
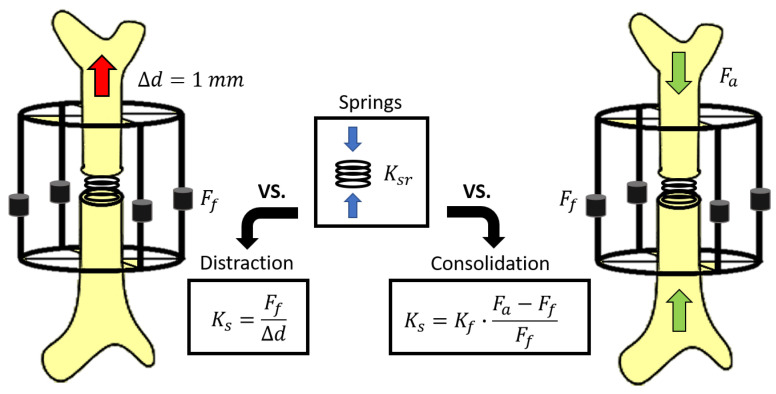
Scheme of the bone-fixator model to estimate the stiffness of the bone callus during the distraction phase (left) and the consolidation phase (right).

**Figure 10 sensors-20-04591-f010:**
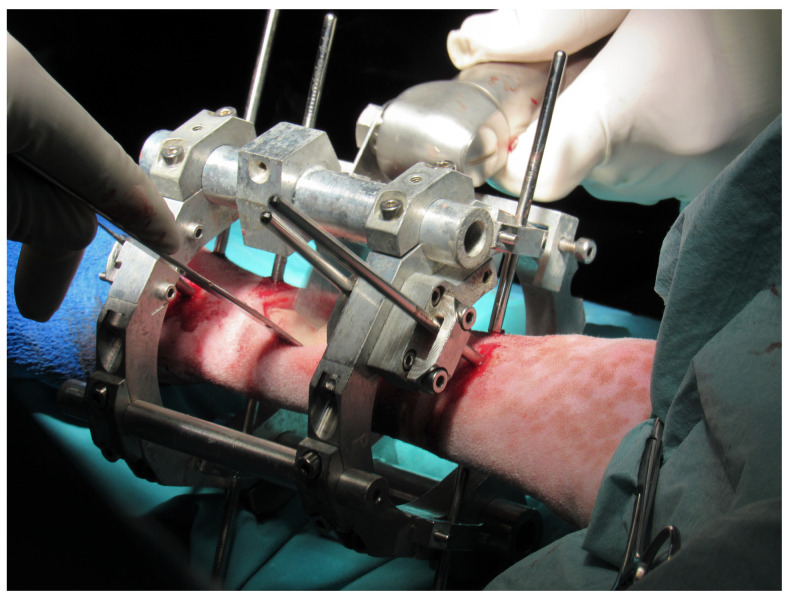
Implantation of the designed external fixator with non-instrumented bars into the sheep metatarsus and performance of the osteotomy, which divides the metatarsus into two bony fragments.

**Figure 11 sensors-20-04591-f011:**
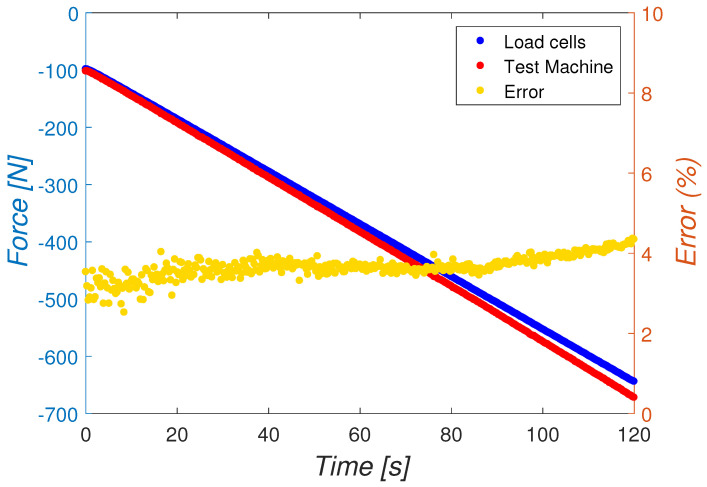
Comparison between forces from compression machine (test machine) and the sum of forces measured by the load cells in a compression test of the empty external fixator.

**Figure 12 sensors-20-04591-f012:**
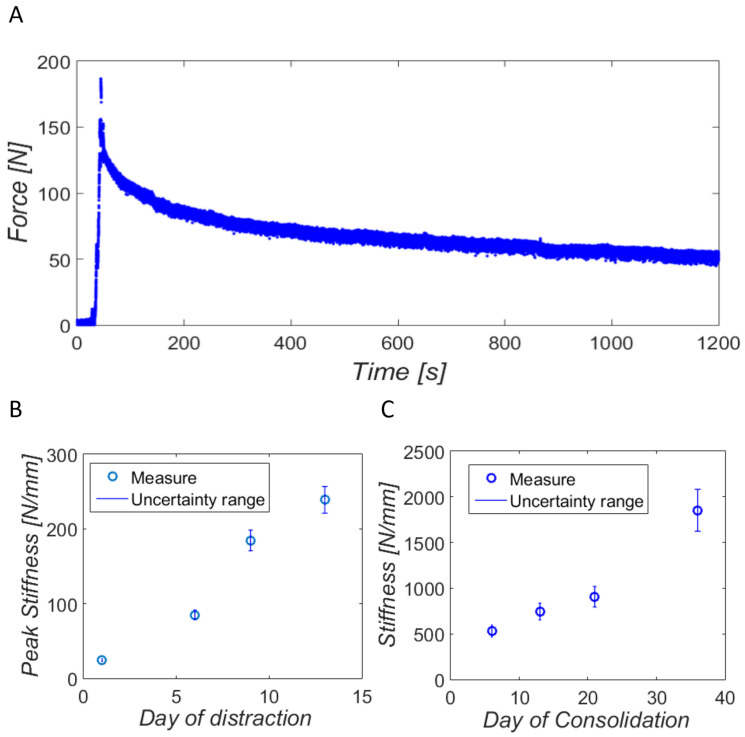
In vivo results: (**A**) distraction force measurement the ninth day of distraction of our specimen; (**B**) in vivo estimation of the peak stiffness of the bone callus at different time-points of the distraction phase and their uncertainty range; (**C**) estimation of the stiffness of the bone callus at different time-points of the consolidation phase and their uncertainty range.

**Table 1 sensors-20-04591-t001:** Results from the calibration test. $K_{sr}$: stiffness of the elastic-spring. $K_{s}$: stiffness estimated with the bone-fixator model. $\sigma_{s}$: standard deviation of the stiffness estimations e: error in the estimated stiffness value. U: Uncertainty from reference, repetition and replication errors.

Distraction					Consolidation				
**$K_{\mathrm{sr}}$ (N/mm)**	**$K_{s}$ (N/mm)**	**$\sigma_{s}$ (N/mm)**	**e (%)**	**U (%)**	**$K_{\mathrm{sr}}$ (N/mm)**	**$K_{s}$ (N/mm)**	**$\sigma_{s}$ (N/mm)**	**e (%)**	**U (%)**
16.75	17.25	1.20	3.01	8.91	103.01	98.79	16.52	4.09	21.92
39.93	35.65	2.86	10.72	10.74	208.83	194.29	14.62	6.96	9.65
65.45	62.88	4.03	3.93	8.29	416.37	398.33	31.86	4.33	14.22
102.77	92.15	3.21	10.33	5.01	1979.03	1960.38	148.25	0.94	12.15
175.01	167.09	4.90	4.52	3.81	5050.13	4572.29	539.96	9.46	16.02
208.83	192.39	9.03	7.87	8.26	7448.78	7716.09	567.94	3.59	10.28
**average**			**6.73**	**7.50**	**average**			**4.90**	**14.04**

**Table 2 sensors-20-04591-t002:** Acquisition systems for in vivo monitoring of bone regeneration processes.

Study	Hardware/Software	Real-Time	Size (mm)	Weight (g)	Portable	2nd Storage
Grasa et al. [27]	N.D./Specific	Yes	150 × 100 × 45	>315	Yes	No
Mora-Macías et al. [30]	Commercial	Yes	250 × 200 × 65	2430	Yes	No
Reifenrath et al. [34]	Commercial	Yes	120 × 80 × 55	420	No	No
Meyers et al. [36]	Commercial	Yes	N.D.	N.D.	No	No
Wee et al. [47]	Commercial	No	15 Ø × 38 L	30	Yes	N.D.
This work	Specific	Yes	125 × 80 × 33	173	Yes	Yes

N.D. The article does not provide data.

**Table 3 sensors-20-04591-t003:** Medical devices for bone regeneration applications that reported error data in their measures.

Study	Measurement	Errors
Mora-Macías et al. [30]	Bone callus axial stiffness in bone transport	7.8%/9.5% *
Widhagen et al. [57]	Bone callus torsional stiffness in distraction osteogenesis	∼15%
Hente et al. [58]	Bending stiffness in fracture healing	∼29.3%
Eastaugh-Waring et al. [59]	Tissue stiffness in fracture healing	∼10%
This work	Tissue axial stiffness in several processes	6.7%/4.9% *

* Errors in bone callus stiffness measures during distraction and consolidation phases, respectively.

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
