# Peer review of "Real-Time Wireless Platform for In Vivo Monitoring of Bone Regeneration"

_sensors, 2020, doi:10.3390/s20164591_

Round 1
Reviewer 1 Report
The authors have instrumented a versatile external fixator capable of adapting to multiple bone regeneration processes. The work is supported by solid implementation but somehow lacks the depths required for showing the clinical efficacy. The authors have criticized other available devices but have not clearly indicated the clinical gains in using the device beyond using some fuzzy qualitative matrices as in Table 2. The authors are requested to define specific test protocols and obtain real user and usability engineering data to quantitatively measure the improvement from a usability perspective rather showing multiple small yet accumulated progresses like adding SD card.
More specifically, the authors are requested to improve the presentation and English language in the text. They should test and benchmark the findings against literature values in quantitative ways.
The novelty factor needs to be clearly stated and important gains and pitfalls need to be univocally stated. The current version of the manuscript looks more like a project implementation report, the authors should highlight the progress in state-of-the-art made and its clinical applications instead.
Reviewer 2 Report
The manuscript is well written and should be accepted for publication.
Reviewer 3 Report
The manuscript entitled "Real Time Wireless Platform for In vivo Monitoring
of Bone Regeneration" reports the development and test of a new external fixator for mechanical monitoring of bone regeneration processes.
The manuscript is interesting, well organized and written. However, I have some concern for publication.
In the aim of the study, Authors reports that the device "allows performing a mechanical monitoring of any bone regeneration process." Probably, not any regeneration process can be monitored by this instrument. As reported in discussion, it could be applied to long bones regeneration but it seems difficult its application for oral or cranial bones even if their assessment would be interesting. Aim should be modified. Furthermore, it would be interesting the discussion of bone healing model that would be investigated by this method.
line 316: it is reported: "the absence of much surrounding soft tissue and its similarity to human limb bones." A reference should be added for this sentence.
Reviewer 4 Report
Review Sensors – 869944
The manuscript entitled “Real Time Wireless Platform for In vivo Monitoring of Bone Regeneration” consists in a very well designed study reporting the development and proof-of-concept of novel versatile system for the in vivo mechanical characterization of bone regeneration processes. The manuscript is well organized and fits perfectly in the scope of the journal Sensors (ISSN 1424-8220). Thus, I strongly recommend its publication in the journal Sensors after the revision of few minor issues.
- Page 1, lines 23-24: MSCs are considered bone progenitor cells, since they have the ability to differentiate into osteoblasts, which are the main bone-producing cells. The authors should rephrase the sentence and elaborate a little more on this part about the effects of mechanical environment on the differentiation and function of MSCs.
- Page 13, lines 362-365: The authors should provide a little more detailed analysis (using appropriate references) about the biological factors involved in the increase of bone stiffness observed. This can be done in this section or separately in the Discussion section.
- Because it is consistently referred throughout the manuscript and of its possible huge impact in the emerging field of regenerative medicine, the authors should provide a more detailed example of a possible future application of the developed device in a bone tissue engineering approach.
Reviewer 5 Report
Technical
- The topic is appropriate for the journal.
- The work has a very clear structure.
- All the ideas are clearly and concisely expressed, and the concepts are understandable. The sections are well written in a way that is easy to read and understand.
- The overall balance and structure of the paper is good. Moreover, all the sections are necessary and properly written.
- English language seems to be appropriate.
Quality
- The paper deals with a Real Time Wireless Platform for In vivo Monitoring of Bone Regeneration, reporting interesting results. In all the manuscript the authors discuss about bone regeneration as well as bone tissue engineering. In addition, in the introduction of the work the authors start to state: “Regeneration processes are intrinsic mechanisms in bone tissue which commonly appear along human lifetime, during fracture healing, bone remodeling or during growth in children. Depending on the process, bone regeneration may have different goals: to recover the skeletal functions of the body, to renew bone tissue or to repair defects. Bone regeneration is carried out by means of multiple key factors, not only biological ones are important [1], but also the mechanical environment plays a remarkable role in the regulation of the activities of mesenchymal stem cells (MSC), main source of bone tissue [2].” Accordingly, the authors should BRIEFLY report some concepts and progresses in the design of advanced devices and systems for bone regeneration (e., “Systematic analysis of injectable materials and 3D rapid prototyped magnetic scaffolds: from CNS applications to soft and hard tissue repair/regeneration. Procedia Engineering. 2013, 59:233–239. Then, the authors should continue to stress their study related to a Real Time Wireless Platform for In vivo Monitoring of Bone Regeneration. All of this should improve the quality of the paper, reporting the progresses in the design of advanced devices and systems as well as in understanding and improving the mechanism of monitoring in the case of bone regeneration. This should clearly help the potential readers to better understand the content of their work.
- The approach is interesting
- The introduction should be improved.
- The list of references should be improved.
Presentation
- The quality of some figures should be improved.
- The title is adequate and appropriate for the content of the article.
- The abstract contains information of the article.
- Figures and captions are essential and clearly reported.
Round 2
Reviewer 1 Report
The overall paper quality has improved, and the responses to review comments are satisfactory. However, the additional details as included in responses to reviewers are not suitably reflected in the main manuscript.
Kindly make sure the relevant details and table (Table R1.1) /figure is suitably added to the main manuscript.
